# Rare Entity Prediction: Language Understanding with External Knowledge using Hierarchical LSTMs

## Abstract

Reading comprehension in NLP refers to the ability of models to answer any question about a passage accurately. An important open problem is how to effectively use external knowledge to answer such questions. In this paper, we introduce a new task and derive new models to drive progress towards this goal. In particular, we propose the task of *rare entity prediction*: given a web document with several entities removed, models are tasked with predicting the correct missing entities conditioned on the document context and the lexical resources. This task is challenging due to the diversity of language styles and the extremely large number of rare entities. Our experiments show that models that make use of external knowledge in the form of lexical resources, particularly our model using hierarchical LSTMs, perform significantly better at rare entity prediction than those that do not.

## 1 Introduction

One of the preeminent problems in natural language processing is the design of models that can acquire knowledge of the world through language. Such knowledge can be obtained in at least two ways: through *unstructured texts* such as news articles or blogs, and through *structured knowledge bases* such as WordNet (Miller, 1995) and Freebase (Bollacker et al., 2008).

A natural setting for testing the abilities of models to acquire knowledge through natural language is *reading comprehension*, where models must answer questions about a given document. Many existing reading comprehension tasks, however, can be solved using basic language modelling, and re-

| Context |
| --- |
| [...] ______, who lived from 1757 to 1827, was admired by a small group of intellectuals and artists in his day, but never gained general recognition as either a poet or painter. [...] |

| Candidate Entities |
| --- |
| Peter Ackroyd: Peter Ackroyd is an English biographer, novelist and critic with a particular interest in the history and culture of London. [...] |
| William Blake: William Blake was an English poet, painter, and printmaker. [...] |
| Emanuel Swedenborg: Emanuel Swedenborg was a Swedish scientist, philosopher, theologian, revelator, and mystic. [...] |

Table 1: An abbreviated example from the Wikilinks Entity Prediction dataset. Shown is an excerpt from the text (context), with a missing entity that must be predicted from a list of candidate entities. Each candidate entity is also provided with its description from Freebase. Due to the large number of rare entities, models must often learn to effectively leverage this external knowledge.

quire little reasoning. This is the case for the Daily Mail/ CNN dataset (Hermann et al., 2015), for example, which has been shown to have a small gap between machine and human performance (Chen et al., 2016). Existing reading comprehension tasks target reasoning about generic concepts, while accounting for syntax, lexical semantics, and/or discourse. In this work, we aim to move towards reasoning about specific instances of entities in context. This is a very difficult problem, as we have very few training samples per instance; thus, we demonstrate that we cannot simply rely on language modelling, and must leverage external sources of knowledge.

Recent efforts have been made to integrate different sources of knowledge, for example com-

bining distributional and relational semantics for building word embeddings (Fried and Duh, 2014; Long et al., 2016). These models show performance improvements on tasks such as knowledge base completion. However, such ideas are rarely examined for reading comprehension tasks.

In this paper, we propose a novel task called *rare entity prediction*, where models must predict missing entities (e.g. proper names or places) in web articles by leveraging the available lexical resources. Using lexical resources is necessary as there are a large number of rare entities in the dataset (see Table 2); thus, it is much more difficult to obtain information about these entities using the unstructured texts alone. For this task, we provide a significantly enhanced version of the Wikilinks dataset, with entity descriptions extracted from Freebase serving as the lexical resources, which we call the *Wikilinks Entity Prediction* dataset. An example from the Wikilinks Entity Prediction dataset is shown in Table 1.

In addition to the task and dataset, we introduce several models that can be used to solve this task, using distributional semantics and recurrent neural networks. We show that language modelling baseline models that do not consider the Freebase descriptions are unable to achieve good performance on the task. RNN models leveraging the external knowledge perform much better; in particular, our hierarchical double encoder model (HIERENC) achieves a 17% increase in accuracy over the language model baseline.

Rare entity prediction task is unique because it is difficult for models that heavily rely on word co-occurrence statistics to achieve good performance. To solve the task, models need to have the abilities to incorporate external a priori knowledge with an understanding of the unstructured natural language. Some questions can be challenging to answer even for humans without proper prior knowledge about the candidate entities. Table 1 provides a good example of such scenarios. We believe that being able to integrate multiple sources of knowledge is crucial for not only reading comprehension, but also other NLP systems.

## 2   Related Work

Related to our work is the task of *entity prediction*, also called *link prediction* or *knowledge base completion*, in the domain of multi-relational data. Multi-relational datasets like WordNet (Miller, 1995) and Freebase (Bollacker et al., 2008) consist of entity-relation *triples* of the form (head, relation, tail), indicating a relationship between the head and tail entity as described by the relation. In entity prediction, either the head or tail entity is removed, and the model has to predict the missing entity. While this task necessitates understanding structured information in the form of relationships between objects, it does not require the processing of natural language text, which is crucial in rare entity prediction.

Also related to our approach is the task of *script learning* or *script induction*, in which a system is asked to predict an event, or a (verb, dependency) pair, that is held out from a document (Chambers and Jurafsky, 2008, 2009; Jans et al., 2012). For example, the event '$X$ orders $Y$' could predict the event '$X$ eats $Y$'. Approaches to this problem have used relational atoms similar to relation triples to represent events (Pichotta and Mooney, 2014), or have used recurrent neural networks (Pichotta and Mooney, 2016). Similarly, the task of *discourse coherence modeling* involves determining text relatedness at different levels of granularity (Grosz et al., 1995). While these tasks involve understanding unstructured natural language, and there are entity-based approaches for solving them (Barzilay and Lapata, 2008), they differ from our task as they do not explicitly require the use of structured knowledge sources such as Freebase. Events can be represented, for example, as relational atoms; for example, the event '$X$ eats $Y$' could be represented as $eat(X, Y)$.

Rare entity prediction is clearly distinct from tasks such as entity tagging and recognition (Ritter et al., 2011), as models are provided with the actual name of the entity in question, and only have to match the entity with related concepts and tags. It is more closely related to the machine reading literature from e.g. Etzioni et al. (2006); however, the authors define machine reading as primarily unsupervised, whereas our task is supervised.

A similar supervised reading comprehension task was proposed by Hermann et al. (2015) using news articles from CNN and the Daily Mail. Given an article, models are tasked with filling in blanks of one-sentence summaries of the article. The original dataset was found to have a low ceiling for machine improvement (Chen et al., 2016); thus, alternative datasets have been proposed that consist of more difficult questions (Trischler et al.,

2016; Rajpurkar et al., 2016). A dataset with a similar task was also proposed by Hill et al. (2015a), where models must answer questions about short children's stories. While these tasks require the understanding of unstructured natural language, it does not require integration with external knowledge sources.

Hill et al. (2015b) propose a method of combining distributional semantics with an external knowledge source in the form of dictionary definitions. The purpose of their model is to obtain more accurate word and phrase embeddings by combining lexical and phrasal semantics, and they achieve fairly good performance on a reverse dictionary and crossword QA tasks.

Perhaps the most related approach to our work is the one concurrently developed by Ahn et al. (2016). The authors propose a WikiFacts dataset where Wikipedia descriptions are aligned with Freebase facts. While they also aim to integrate external knowledge with unstructured natural language, their task differs from ours in that it is primarily a language modeling problem.

Finally, methods that address factoid question answering (QA) using Freebase (Bordes et al., 2014a; Yao and Van Durme, 2014; Serban et al., 2016) must also combine knowledge from a structured knowledge base with an understanding of the question that's being asked. However, the nature of the unstructured data is different: in factoid QA, questions are often short and relate directly to the relations of a given entity, while rare entity prediction requires learning much longer-term dependencies in natural language, where relations between entities are not specifically referred to.

## 3 Rare Entity Prediction

### 3.1 The *Wikilinks* Dataset

The *Wikilinks* dataset (Singh et al., 2012) is a large dataset originally designed for cross-document coreference resolution, the task of grouping entity mentions from a set of documents into clusters that represent a single entity. The dataset consists of a list of non-Wikipedia web pages (discovered using the Google search index) that contain hyperlinks to Wikipedia, such as random blog posts or news articles. Every token with a hyperlink to Wikipedia is then marked and considered an entity mention in the dataset. Each entity mention is also linked back to a knowledge base through their corresponding Freebase IDs.

| Number of documents | 269,469 |
|---|---|
| Average # blanks *per* doc | 3.69 |
| Average # candidates *per* doc | 3.35 |
| Number of unique entities | 245,116 |
| # entities with $n <= 5$ | 207,435 (**84.6%**) |
| # entities with $n <= 10$ | 227,481 (**92.8%**) |
| # entities with $n <= 20$ | 238,025 (**97.1%**) |

Table 2: Statistics for the augmented version of the *Wikilinks* dataset, where $n$ represents the entity frequency in the corpus. Web pages with more than 10 blanks to fill are filtered out for computational reasons.

In order to ensure the hyperlinks refer to the correct Wikipedia pages, additional filtering is conducted to ensure that either (1) at least one token in the hyperlink (or *anchor*) matches a token in the title of the Wikipedia page, or (2) the anchor text matches exactly an anchor from the Wikipedia page text, which can be considered an alias of the page. As many near-duplicate copies of Wikipedia pages can be found online, any web pages where more than 70% of the sentences match those from their linked Wikipedia pages are discarded. To ensure the quality of data, the authors performed manual inspection on a small subset of the web pages. No incorrect hyperlinks were found.

### 3.2 The *Wikilinks Rare Entity Prediction* Dataset

We provide a significantly pre-processed and augmented version of the *Wikilinks* dataset for the purpose of entity prediction, which we called the *Wikilinks Rare Entity Prediction* dataset[1]. In particular, we parse the HTML texts of the web pages and extract their page contents to form our corpus. Entity mentions with hyperlinks to Wikipedia are marked and replaced by a special token (**\*\*blank\*\***), serving as the placeholder for missing entities that we would like the models to predict. The correct missing entity $\tilde{e}$ is preserved as a target. Additionally, we extract the lexical definitions of all entities that are marked in the corpus from Freebase using their Freebase IDs, which are available for all entities in the *Wikilinks* dataset. These lexical definitions will serve as the external knowledge to our models.

Table 2 above shows some basic statistics of our dataset. As we can see, unlike the Children's

---

[1] We will make this dataset publicly available upon acceptance of the paper.

[...] Sinclair doesn't like it, but admits that such change is a constant in London 's history. This book is the transcript of a talk that Sinclair gave to the Swedenborg Society in 2007, and begins with a reflection on how London is being re-shaped in preparation for the 2012 Olympic Games. Blake sensed these ancient presences in London and the power and energy they generated in the life of the city. *Like other London writers such as Will Self and **blank**, Sinclair is an avid walker about the city and its surrounds, and an absorbed reader of the palimpsest that is the modern capi-tal.* It is almost impossible to walk anywhere in London and not be drawn into the past lives, buildings and cultures that have driven its existence. Ancient and modern, and all steps in between , lie in the city 's topography, some of it visible and some long buried. Sinclair recalls a visit to London in 1965 by the American poet Allen Ginsberg and how they were both inspired by the works of Blake. [...]

---

**Context**

*Like other London writers such as Will Self and* $\underset{w_1 \quad w_2}{} \quad ... \quad \underset{w_{blank}}{**blank**}$,

*Sinclair is an avid walker about the city and its surrounds, and*
...

*an absorbed reader of the palimpsest that is the modern capital.*
$\qquad\qquad\qquad ... \qquad w_{n-1} \quad w_n$

---

**Candidate Entities**

Peter Ackroyd: Peter Ackroyd is an English biographer, novelist and $\big\}\, l_1$
$\underset{e_1}{} \qquad \underset{l_{1,1}}{} \quad \underset{l_{1,2}}{} \qquad ...$
critic with a particular interest in the history and culture of London.
$\qquad\qquad\qquad\qquad\qquad ... \qquad\qquad l_{1,k_1-1} \quad l_{1,k_1}$

William Blake: William Blake was an English poet, painter, and printmaker. $\big\}\, l_2$
$\underset{e_2}{} \qquad\qquad l_{2,1} \quad l_{2,2} \qquad\qquad ... \qquad\qquad l_{2,k_2-1} \quad l_{2,k_2}$

Emanuel Swedenborg: Emanuel Swedenborg was a Swedish scientist,
$\underset{e_3}{} \qquad\qquad\qquad l_{3,1} \quad l_{3,2} \qquad ...$
philosopher, theologian, revelator, and mystic. $\big\}\, l_3$
$\qquad ... \qquad\qquad l_{3,k_3-1} \quad l_{3,k_3}$

Figure 1: An example from the *Wikilinks Rare Entity Prediction* dataset. Shown is a paragraph from the dataset, along with the context (in blue italics) and the missing entity (in red underline). We also visually show the notation that we use for the remainder of this paper. The correct answer to this question is Peter Ackroyd.

Book dataset, which has 50k candidate entities for almost 700k context and query pairs (Hill et al., 2015a), the number of unique entities found in our dataset is in the same order of magnitude as the number of documents available.

Moreover, the majority of entities appears a rel-atively small number of times, with 92.8% ob-served less than or equal to 10 times across the entire corpus. This suggests that models that only rely on the surrounding context information may not be able to correctly predict the missing enti-ties. This further motivates us to incorporate ad-ditional information into the decision process to improve the performance. In the experiments sec-tion, we show that the lexical resources are indeed necessary to achieve better results.

### 3.3 Task Definition[2]

Here, we formalize the task definition of the en-tity prediction problem. Given a document $\mathcal{D}$ in the corpus, we split it into an ordered list of con-texts $\mathcal{C} = \{\mathsf{C}_1, ..., \mathsf{C}_n\}$ where each context $\mathsf{C}_i$ $(1 \leq i \leq n)$ is a single sentence $(w_1, ..., w_m)$ where the special token **blank** is found.[3] Let $\mathcal{E}$ be the set of candidate entities. For each miss-ing entity, we want the model to select the correct entity $e \in \mathcal{E}$ to fill the blank slot. In our problem

setting, the model also has access to the lexical re-source $\mathcal{L} = \{\mathsf{L}_e \mid e \in \mathcal{E}\}$ where $\mathsf{L}_e = (l_{e1}, ..., l_{ek})$ is the lexical definition of entity $e$ extracted from the knowledge base. Thus, the task of the model is to predict the correct missing entities for each empty slot in $\mathcal{D}$.

There are several possible ways to specify the candidate set $\mathcal{E}$. For instance, we could define $\mathcal{E}$ so that it includes all entities found in the cor-pus. However, given the extremely large amount of unique entities found in the dataset, this would render the task difficult to solve from both a prac-tical and computational perspective. We present a simpler version of the task where $\mathcal{E}$ is the set of entities that are present in the document $\mathcal{D}$. Note that we can make the task arbitrarily more difficult by randomly sampling other entities from the en-tity vocabulary and adding them to the candidate set.

We show an example from the *Wikilinks Entity Prediction* dataset, along with a visual guide to the notation from this section, in Figure 1.

## 4 Model Architectures

In this section, we present two models that use the lexical definitions of entities to solve the pro-posed rare entity prediction problem. The basic building blocks of our models are recurrent neu-ral networks (RNN) with long short-term mem-ory (LSTM) units. An RNN is a neural network with feedback connections that allows information from the past to be encoded in the hidden layer

---

[2] On notation; we use A to denote sequences, $\mathcal{A}$ to denote sets, $a$ to denote words/entities, $\mathbf{a}$ to denote vectors, $A$ to denote matrices.

[3] Note that we restrict each context to only have one **blank** token.

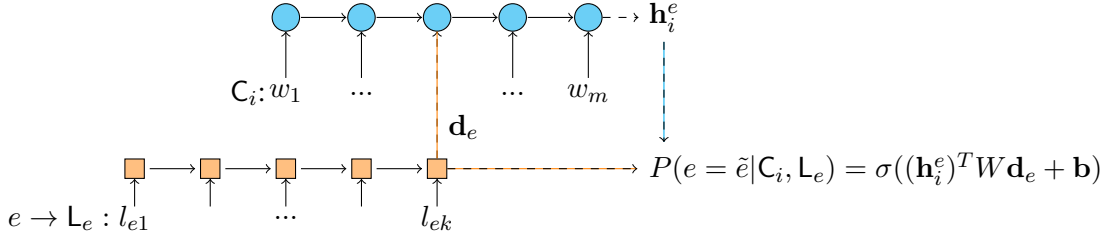

Figure 2: Our double encoder architecture. Each entity $e$ has an associated lexical definition $\mathsf{L}_e = (l_{e1}, l_{e2}..., l_{ek})$, which is fed through the lexical encoder $g$ (orange squares) to provide an encoding $\mathbf{d}_e$. This definition embedding is then fed as the *blank* input token of context $\mathsf{C}_i$ to the context encoder $f$ (blue circles), which provides a context embedding $\mathbf{h}_i^e$.

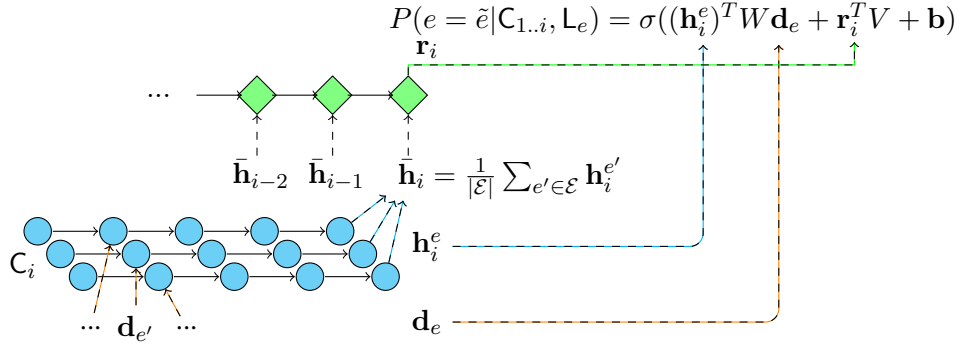

Figure 3: Our hierarchical encoder architecture. Each entity $e$ is encoded as $\mathbf{d}_e$, at each timestep, $\mathbf{h}_i^e$ is computed for each $e$. $\bar{\mathbf{h}}_i$ is the average encoding, which is fed as input to the temporal model $r$ (green diamonds). The temporal model produces $\mathbf{r}_i$, which is used to compute $P(e = \tilde{e}|\mathsf{C}_{1..i}, \mathsf{L}_e)$.

representation, thus is ideal for modelling sequential data (Dieterich, 2002) and most language related problems.

LSTMs are an extension of RNNs which include a memory cell $\mathbf{c}_t$ alongside their hidden state representation $\mathbf{h}_t$. Reads and writes to the memory cell are controlled by a set of three gates that allow the model to either keep or discard information from the past and update their state with the current input (Hochreiter and Schmidhuber, 1997). This allows LSTMs to model potentially longer dependencies and at the same time mitigate the vanishing and exploding gradient problems, which are quite common among regular RNNs (Hochreiter, 1991; Bengio et al., 1994). In our experiments, we use LSTMs augmented with peephole connections (Gers et al., 2002).

We denote the output (i.e. the last hidden state) of an RNN $f$ operating on a sequence $\mathsf{S}$ as $f(\mathsf{S})$, and subscript the $t$-th hidden state as $f_t(\mathsf{S})$.

### 4.1 Double Encoder (DOUBENC)

We use two jointly trained recurrent models, a lexical encoder $g(.)$ and a context encoder $f(.)$, and a logistic predictor $P$ (see Figure 2).

The lexical encoder converts the definition of an entity into a vector embedding, while the context encoder repeats the same process for a given context to obtain its context embedding. These two embeddings are then used by $P$ to predict if the given entity-context pair is correct. Additionally, the blank in the context sentence is replaced by the encoded definition embedding to provide more information to $f$.

For an entity $e$ in the candidate set $\mathcal{E}$ of document $\mathcal{D}$, we retrieve its corresponding lexical definition $\mathsf{L}_e$, itself a sequence of words, to compute its encoding $g(\mathsf{L}_e) \equiv \mathbf{d}_e$.

For a given context $\mathsf{C}_i$, we replace the embedding of the blank token with $\mathbf{d}_e$. Thus $\mathsf{C}_i = (w_1, ..., w_{blank}, ..., w_m)$ becomes $\mathsf{C}_i^e = (w_1, ..., \mathbf{d}_e, ..., w_m)$[4]. We then compute the context embedding of the new $\mathsf{C}_i^e$, $f(\mathsf{C}_i^e) \equiv \mathbf{h}_i^e$

After getting $\mathbf{h}_i^e$ and $\mathbf{d}_e$, we wish to compute the probability of candidate $e$ being the correct entity

---

[4]We mix the word/vector notation here since each word $w$ is replaced by its corresponding word embedding vector.

$\tilde{e}$ missing in context $\mathsf{C}_i$. This probability is the output of the predictor:

$$P(e = \tilde{e}|\mathsf{C}_i, \mathsf{L}_e) = \sigma((\mathbf{h}_i^e)^T W \mathbf{d}_e + \mathbf{b})$$

where $\sigma$ is the sigmoid function, $W$ and $\mathbf{b}$ are model parameters. The cross term $(\mathbf{h}_i^e)^T W \mathbf{d}_e$ is a dot product between $\mathbf{h}_i^e$ and $\mathbf{d}_e$ that weighs the dimensions differently based on the learned parameters $W$. A similar prediction method has been used successfully for question answering (Bordes et al., 2014b; Yu et al., 2014) and dialogue response retrieval (Lowe et al., 2015). Note that while $\mathbf{h}_i^e$ is a function of $\mathbf{d}_e$, using $\mathbf{d}_e$ in the cross term $(\mathbf{h}_i^e)^T W \mathbf{d}_e$ provides a much shorter gradient path from the loss to the lexical encoder through $\mathbf{d}_e$.

Given a context, the model outputs a probability for each entity $e \in \mathcal{E}$. Entities in the candidate set are then ranked against each other according to their predicted probabilities. The entity with the highest probability is considered as the most plausible answer for the missing entity in the current context. We consider the model to make an error if that entity is not $\tilde{e}$.

## 4.2 Hierarchical Double Encoder (HIERENC)

The double encoder model above considers each context independently. However, since each document consists of a sequence of contexts, the knowledge carried by other contexts in $\mathcal{C}$ could also provide useful information for the decision process of $\mathsf{C}_i$.

To that end, we propose a hierarchical model structure by adding a LSTM network, which we call the temporal network $r$ (see Figure 3), on top of the double encoder architecture. Since a document is a sequence of $\mathsf{C}_i$s, each *timestep* of this network consists of one such context, and thus is indexed with $i$.

Since we already have a context encoder $f$, we reuse the output of $f(\mathsf{C}_i^e)$ as the input of $r$ at timestep $i$. More specifically, we combine the embeddings generated by $f$ into a single one via averaging: $\bar{\mathbf{h}}_i = \frac{1}{|\mathcal{E}|} \sum_{e' \in \mathcal{E}} \mathbf{h}_i^{e'}$, which then serves as the input to the temporal network for context $\mathsf{C}_i$. Note that alternatively, one could aggregate information about the past predictions through other means like policies or soft attention. However, this would introduce extra complexities to the learning process. As such, we use averaging to that end.

Finally, at each timestep $i$, the temporal network outputs an embedding $r_i(\mathsf{C}_1, ..., \mathsf{C}_n) \equiv \mathbf{r}_i$. We use

this temporal embedding to predict the probability of the context-entity pair using slightly altered $P$:

$$P(e = \tilde{e}|\mathsf{C}_{1..i}, \mathsf{L}_e) = \sigma((\mathbf{h}_i^e)^T W \mathbf{d}_e + \mathbf{r}_i^T V + \mathbf{b})$$

where $W$, $V$ and $\mathbf{b}$ are model parameters. The entities in candidate set are again ranked against each other based on their probabilities.

## 5 Experiments

### 5.1 Experiment Setup

We randomly partition our dataset into training, validation and test sets. The training set consists of approximately 80% of the total dataset, the validation and test sets comprise about 10% each.

In our experiments, the contexts are defined as the sentences where the special **blank** tokens are found. The lexical definitions for each entity are the first sentences of their Freebase descriptions. We have experimented with different configurations of defining contexts and entity definitions, such as expanding the context window by including the sentences before and after the blank, as well as taking more than one sentence out of the entity description. However, results on the validation set show that increasing the context window size and definition size has very little impact on the evaluation metrics and drastically increases the training time of all models.

To train our models, we use the correct missing entity for each blank as the positive sample and all other entities in the candidate set as the negative samples. During the testing phase, we present each entity in the candidate set to our models and record the probabilities output by the models. The entity with the highest probability is chosen as the model prediction. For all gradient-based methods, including both baseline models and our proposed models, the learning objective is to minimize the binary cross-entropy of the training data.

We measure the performance on our entity prediction task using the *accuracy*; that is, the number of correct entity predictions made by the model divided by the total number of predictions. This is equivalent to the metric of Recall@1 that is often used in information retrieval. All our models are implemented using Theano (Theano Development Team, 2016).

### 5.2 Baselines

In order to demonstrate the effects of using lexical resources as external knowledge for solving

the task, we present here three sets of baselines: one set of simple baselines (RANDOM and FREQENT), one LSTM-based model that only relies on the contexts and does not utilize the definitions (CONTENC), and another set of models that do make use of the entity definitions but in a relatively simple fashion (TF-IDF + COS and AVGEMB + COS).

**RANDOM** For each context in a given document, this baseline simply selects an entity from the candidate set uniformly at random as its prediction.

**FREQENT** Under this baseline, we rank all entities in the candidate set by the number of times that they appear in the document. For each blank in the document, we always choose the entity with the highest frequency in that document as the prediction. Note that this model has access to extra information compared to the other models, in particular the total number of times each entity appears in the document.

**CONTENC** Instead of using their definitions, entities are treated as regular tokens in our vocabulary. Thus for a particular entity $e$, the context sequence $\mathsf{C}_i = [w_1, ..., w_{blank}, ..., w_m]$ becomes $[w_1, ..., w_e, ..., w_m]$. We feed the sequence $\mathsf{C}_i$ into the context encoder and as usual take the last hidden state as the context embedding $\mathbf{h}_i^e$. Thus given $\mathsf{C}_i$ and $e \in E$, the probability of $e$ being the correct missing entity is:

$$P(e = \tilde{e}|\mathsf{C}_i) = \sigma((h_i^e)^T W + \mathbf{b})$$

where again $\sigma$ is the sigmoid function, $W$ and $\mathbf{b}$ are model parameters. This model is essentially a language model baseline, that does not make use of the external a priori knowledge.

**TF-IDF + COS** This method takes the term frequency-inverse document frequency (TF-IDF) vectors of the context and the entity definition as their corresponding embeddings. The aggregations of contexts and definitions are treated as their own corpora, and two separate TF-IDF transformers are fitted. Candidate entities are ranked by the *cosine similarity* between their definition vectors and the context vector. The entity with the highest cosine similarity score is chosen as the prediction.

**AVGEMB + COS** This baseline computes the context embedding by taking the *average* of some

| | Accuracy | |
|---|---|---|
| **Model** | Valid | Test |
| **Fixed baselines** | | |
| RANDOM | 0.294 | 0.301 |
| FREQENT | 0.329 | 0.331 |
| **Without external knowledge** | | |
| CONTENC | 0.393 | 0.396 |
| **With external knowledge** | | |
| TF-IDF + COS | 0.292 | 0.300 |
| AVGEMB + COS | 0.355 | 0.359 |
| DOUBENC | 0.547 | 0.540 |
| HIERENC | **0.573** | **0.566** |

Table 3: Results on the Wikilinks Entity Prediction dataset for all proposed baselines and models.

pre-trained word embeddings. The entities' embeddings are computed in the same way. In our experiments, we choose to use the published *GloVe* (Pennington et al., 2014) pre-trained word embeddings. Same as above, the prediction is made by considering the cosine similarity between the context embedding and the entity embeddings.

## 5.3 Results

Empirical results are shown in Table 3. We test our two proposed model architectures (detailed in Section 4), in addition to baselines described in Section 5.2.

For the CONTENC baseline, we choose 300 as the size of hidden state for the encoder. For the DOUBENC and the HIERENC models, the size of hidden state for both the context encoder and the lexical encoder is set to 300. Additionally, an RNN with 200 LSTM units is used as the temporal network in the hierarchical case. All three models are trained with stochastic gradient descent with Adam (Kingma and Ba, 2014) as our optimizer, with a learning rate of 0.0005 used for CONTENC and 0.0001 used for DOUBENC as well as HIERENC. Models with the best performance on validation set are saved and used to test on the test set.

It is clear from Table 3 that models that only use contextual knowledge give relatively poor performance compared to the ones that utilize lexical resources. The large discrepancy between the context encoder and the double encoder shows that lexical resources play a crucial role in solving the task. The best result is achieved by the hierarchical double encoder, which confirms that knowing about previous contexts is indeed beneficial to the prediction at the current timestep.

| Context & Prediction |
| --- |
| [...] We heard from Audrey Bomse, who is with the Free Gaza movement. She was in ______, Cyprus. [...] |
| CONTENC: Istanbul      HIERENC: Larnaca |

| Candidate Set |
| --- |
| Istanbul: Istanbul is the most populous city in Turkey, and the country's economic, cultural, and historical center. |
| Larnaca: Larnaca is a city on the southern coast of Cyprus and the capital of the eponymous district. |
| Ben Macintyre: Ben Macintyre is a British author, historian, reviewer and columnist writing for The Times newspaper. |
| *(Other candidates......)* |

Table 4: An example from the test set, with the predictions made by CONTENC and HIERENC. HIERENC was able to successfully predict the correct missing entity, *Larnaca*.

However, even our best performing model, the hierarchical double encoder, is still a long way from solving this task, achieving an accuracy of 56.6%. This reflects the challenging nature of the task, as most of the predictions are about entities that have been seen fewer than 5 times in the entire corpus. Note that, since Singh et al. (2012) manually verified the quality of the original *Wikilinks* dataset, there are virtually no entities that are incorrectly labeled, and thus this is not a significant source of error.

## 6 Discussion

Table 4 shows an example found in the test set, along with the predictions from CONTENC and HIERENC. Although the context encoder was able to identify that the missing entity should be a city, it incorrectly predicted *Istanbul*. This is likely because *Istanbul* appears 86 times in the dataset, whereas *Larnaca* appears only twice. It seems that, although the context encoder was able to derive a strong association between Istanbul and Middle Eastern geolocations, such knowledge was not learned for Larnaca because of the lack of examples. Conversely, the hierarchical double encoder was able to take both the context and the external knowledge into account and successfully predicted the correct missing city.

One interesting observation is the margin of difference in accuracy between the context encoder and the embedding average baseline. The context encoder, which is a relatively sophisticated context-only model, only slightly outperforms the simple embedding average baseline that has no learning component. This suggests that the lexical definitions are valuable in solving such tasks even when it is used in a rather simplistic way.

Another interesting observation is that, in our experiments we found that using a large context window size (including the sentences before and after the sentence where the blank is found) does not have any significant positive impact on the results. This implies that the words that are most informative about the missing entity in the blank are generally found in the vicinity of the blank. It is likely that more sophisticated models will be able to use the surrounding context information more effectively, leading to greater performance increases.

## 7 Conclusions

In this paper, we examine the use of external knowledge in the form of lexical resources for solving reading comprehension problems. Specifically, we propose the problem of rare entity prediction. In our *Wikilinks Rare Entity Prediction* dataset, the majority of the entities have very low frequencies across the entire corpus; thus, models that solely rely on co-occurrence statistics tend to underperform. We show that models leveraging the Freebase descriptions achieve large performance increases, particularly when this information is incorporated intelligently as in our double encoder-based models.

For future work, it would be interesting to examine the effects of other knowledge sources. In this paper, we use entity definitions as the source of external knowledge. However, Freebase also contains other types of valuable information, such as relational information between entities. Thus, one potential direction for future work would be to incorporate the relational information, alongside the lexical definitions, to achieve better results.

We have seen the crucial role that external knowledge plays in solving tasks with many rare entities. Thus we believe that incorporating external knowledge into other NLP systems, such as dialogue agents, should also see similar positive results. We plan to explore the idea of external knowledge integration further in our future research.

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
