# Peer review of "Rare Entity Prediction: Language Understanding with External Knowledge using Hierarchical LSTMs"

_ACL 2017 — decision unknown_

[Official Review · Reviewer 1 · rating 4 · confidence 4]
soundness 3 · originality 4 · clarity 4 · impact 3 · substance 4 · appropriateness 5 · meaningful comparison 3 · presentation format Oral Presentation

- Contents:
This paper proposes a new task, and provides a dataset. The task is to predict
blanked-out named entities from a text with the help of an external
definitional resource, in particular FreeBase. These named entities are
typically rare, that is, they do not appear often in the corpus, such that it
is not possible to train models specifically for each entity. The paper argues
convincingly that this is an important setting to explore. Along with multiple
baselines, two neural network models for the problem are presented that make
use of the external resource, one of which also accumulates evidence across
contexts in the same text. 

- Strengths:

The collection of desiderata for the task is well-chosen to advance the field:
predicting blanked-out named entities, a task that has already shown to be
interesting in the CNN/Daily Mail dataset, but in a way that makes the task
hard for language models; and the focus on rare entities should drive the field
towards more interesting models. 

The collection of baselines is well chosen to show that neither a NN model
without external knowledge nor a simple cosine similarity based model with
external knowledge can do the task well.

The two main models are chosen well.

The text is clear and well argued. 

- Weaknesses:

I was a bit puzzled by the fact that using larger contexts, beyond the
sentences with blanks in them, did not help the models. After all, you were in
a way using additional context in the HierEnc model, which accumulates
knowledge from other contexts. There are two possible explanations: Either the
sentences with blanks in them are across the board more informative for the
task than the sentences without. This is the explanation suggested in the
paper, but it seems a bit unintuitive that this should be the case. Another
possible explanation is that the way that you were using additional context in
HierEnc, using the temporal network, is much more useful than by enlarging
individual contexts C and feeding that larger C into the recurrent network.  Do
you think that that could be what is going on?

- General Discussion:

I particularly like the task and the data that this paper proposes. This setup
can really drive the field forward, I think. This in my mind is the main
contribution.

[Official Review · Reviewer 2 · rating 2 · confidence 5]
soundness 3 · originality 4 · clarity 4 · impact 3 · substance 4 · appropriateness 5 · meaningful comparison 3 · presentation format Oral Presentation

- General Discussion:

The paper deals with the task of predicting missing entities in a given context
using the Freebase definitions of those entities. The authors highlight the
importance of the problem, given that the entities come from a long-tailed
distribution. They use popular sequence encoders to encode the context and the
definitions of candidate entities, and score them based on their similarity
with the context. While it is clear that the task is indeed important, and the
dataset may be useful as a benchmark, the approach has some serious weaknesses
and the evaluation leaves some questions unanswered. 

- Strengths:

The proposed task requires encoding external knowledge, and the associated
dataset may serve as a good benchmark for evaluating hybrid NLU systems.

- Weaknesses:

1) All the models evaluated, except the best performing model (HIERENC), do not
have access to contextual information beyond a sentence. This does not seem
sufficient to predict a missing entity. It is unclear whether any attempts at
coreference and anaphora resolution have been made. It would generally help to
see how well humans perform at the same task.

2) The choice of predictors used in all models is unusual. It is unclear why
similarity between context embedding and the definition of the entity is a good
indicator of the goodness of the entity as a filler.

3) The description of HIERENC is unclear. From what I understand, each input
(h_i) to the temporal network is the average of the representations of all
instantiations of context filled by every possible entity in the vocabulary.
This does not seem to be a good idea since presumably only one of those
instantiations is correct. This would most likely introduce a lot of noise.

4) The results are not very informative. Given that this is a rare entity
prediction problem, it would help to look at type-level accuracies, and 
analyze how the accuracies of the proposed models vary with frequencies of
entities.

- Questions to the authors:

1) An important assumption being made is that d_e are good replacements for
entity embeddings. Was this assumption tested?

2) Have you tried building a classifier that just takes h_i^e as inputs?

I have read the authors' responses. I still think the task+dataset could
benefit from human evaluation. This task can potentially be a good benchmark
for NLU systems, if we know how difficult the task is. The results presented in
the paper are not indicative of this due to the reasons stated above. Hence, I
am not changing my scores.

[Official Review · Reviewer 3 · rating 2 · confidence 4]
soundness 3 · originality 4 · clarity 4 · impact 3 · substance 3 · appropriateness 5 · meaningful comparison 3 · presentation format Poster

- Strengths:
The paper empirically verifies that using external knowledge is a benefit.

- Weaknesses:
Real world NLP applications should utilize external knowledge for making better
predictions. The authors propose Rare Entity prediction task to demonstrate
this is the case. However, the motivation of the task is not fully justified.
Why is this task important? How would real world NLP applications benefit from
this task? The paper lacks a convincing argument for proposing a new task. For
current reading comprehension task, the evidence for a correct answer can be
found in a given text, thus we are interested in learning a model of the world
(i.e causality for example), or a basic reasoning model. Comparing to reading
comprehension, rare entity prediction is rather unrealistic as humans are
terrible with remembering name. The authors mentioned that the task is
difficult due to the large number of rare entities, however challenging tasks
with the same or even more difficult level exist, such as predicting correct
morphological form of a word in morphologically rich languages. Such tasks have
obvious applications in machine translation for example.

- General Discussion:
It would be helpful if the authors characterize the dataset in more details.
From figure 1 and table 4, it seems to me that overlapping entities is an
important feature. There is noway i can predict the **blank** in figure 1 if I
don't see the word London in Peter Ackoyd description. That's being said,
before brutalizing neural networks, it is essential to understand the
characteristic of the data and the cognitive process that searches for the
right answer.

Given the lack of characteristic of the dataset, I find that the baselines are
inappropriate. First of all, the CONTENC is a natural choice at the first sigh.
However as the authors mentioned that candidate entities are rare, the
embeddings of those entities are unrealizable. As a consequence, it is expected
that CONTENC doesn't work well. Would it is fairer if the embeddings are
initialized from pre-trained vectors on massive dataset? One would expect some
sort of similarity between Larnaca and Cyprus in the embedding space and
CONTENC would make a correct prediction in Table 4. What would be the
performance of TF-IDF+COS and AVGEMB+COS if only entities are used to compute
those vectors?

From modeling perspective, I appreciate that the authors chose a sigmoid
predictor that output a numerical score between (0,1). This would help avoiding
normalization over the list of candidates, which are rare and is difficult to
learn reliable weights for those. However, a sidestep technique does exist,
such as Pointer Network. A representation h_i for C_i (*blank* included) can be
computed by an LSTM or BiLSTM, then Pointer Network would give a probabilistic
interpretation p(e_k|C_i) \propto exp(dot(d_{e_k}, h_i)). In my opinion,
Pointer Network would be an appropriate baseline. Another related note: Does
the unbalanced set of negative/positive labels affect the training? During
training, the models make 1 positive prediction while number of negative
predictions is at least 4 times higher?

While I find the task of Rare Entity prediction is unrealistic, having the
dataset, it would be more interesting to learn about the reasoning process that
leads to the right answer such as which set of words the model attends to when
making prediction.